# SE-VisionTransformer: Hybrid Network for Diagnosing Sugarcane Leaf Diseases Based on Attention Mechanism

**DOI:** 10.3390/s23208529

**Published:** 2023-10-17

**Authors:** Cuimin Sun, Xingzhi Zhou, Menghua Zhang, An Qin

**Affiliations:** School of Computer and Electronic Information Engineering, Guangxi University, Nanning 530004, China; zhouxingzhi1123@163.com (X.Z.); zhang_meng_hua@163.com (M.Z.); 2213301042@st.gxu.edu.cn (A.Q.)

**Keywords:** sugarcane disease, SE attention, multi-head self-attention, convolutional neural network

## Abstract

Sugarcane is an important raw material for sugar and chemical production. However, in recent years, various sugarcane diseases have emerged, severely impacting the national economy. To address the issue of identifying diseases in sugarcane leaf sections, this paper proposes the SE-VIT hybrid network. Unlike traditional methods that directly use models for classification, this paper compares threshold, K-means, and support vector machine (SVM) algorithms for extracting leaf lesions from images. Due to SVM’s ability to accurately segment these lesions, it is ultimately selected for the task. The paper introduces the SE attention module into ResNet-18 (CNN), enhancing the learning of inter-channel weights. After the pooling layer, multi-head self-attention (MHSA) is incorporated. Finally, with the inclusion of 2D relative positional encoding, the accuracy is improved by 5.1%, precision by 3.23%, and recall by 5.17%. The SE-VIT hybrid network model achieves an accuracy of 97.26% on the PlantVillage dataset. Additionally, when compared to four existing classical neural network models, SE-VIT demonstrates significantly higher accuracy and precision, reaching 89.57% accuracy. Therefore, the method proposed in this paper can provide technical support for intelligent management of sugarcane plantations and offer insights for addressing plant diseases with limited datasets.

## 1. Introduction

Sugarcane is an important cash crop in the Guangxi Zhuang Autonomous Region of China (between 105°08′ E and 112°04′ E longitude, and 20°54′ N and 26°24′ N latitude). The region has a subtropical monsoon climate, characterized by predominantly hot and humid weather, which leads to severe disease and pest infestations. Different stages of sugarcane growth are susceptible to various diseases. Without proper management, these diseases can significantly hinder the growth and maturity of sugarcane seedlings, resulting in a significant reduction in sugarcane yields [1]. At an average incidence rate of 51.4%, sugarcane stem yield decreased by 24.9%, and sugar content decreased by 0.56%. Indeed, the wide variety of diseases affecting sugarcane often leads farmers to make incorrect judgments based on prior knowledge and subjective experiences when they encounter these diseases. This can result in the improper use of pesticides, further decreasing sugarcane yields and posing a serious threat to food security [2]. This also aligns with the concept proposed by Industry 5.0, which emphasizes a people-centric approach, strives for sustainable development, and aims to protect and respect the environment while maximizing the rational utilization of natural resources [3].

The traditional method of identifying plant leaf diseases relies on human visual observation of characteristics such as color variations, size, shape, texture, lesion patterns, and overall appearance of the affected leaf area to determine the specific disease and its severity [4]. However, this subjective experience-based approach often leads to incorrect diagnoses, which can result in reduced crop yields, as mentioned earlier.

With the advancement of scientific technology, machine learning algorithms have been increasingly used to replace the traditional visual observation of plant leaf diseases. support vector machine (SVM) is a classic machine learning binary classification model. Yigit et al. [5] proposed using an SVM classifier to perform binary classification between diseased and healthy sugarcane leaves under simple backgrounds. By analyzing color variations and texture features of these images, they achieved an accuracy of 92%. However, this method is limited to binary classification and does not extend to multi-class classification. Ratnasari et al. [6] collected RGB color images of diseased sugarcane leaf sections under laboratory conditions with a simple background. They employed SVM to extract features from the diseased leaf regions, specifically using the gray-level co-occurrence matrix as the feature matrix. The method achieved an accuracy of 80%. However, a limitation of this approach is that it does not automatically update the parameters within the algorithm. To address the issue of manual parameter adjustment in SVM algorithms, Zhang et al. [7] proposed the use of a genetic algorithm to automatically optimize the parameters of the kernel function in SVM. This approach aims to achieve better performance by adjusting the parameters automatically. However, a consequence of this method is an increase in time cost. Basavaiah et al. [8] proposed that the decision tree classifier achieves a higher accuracy in classification, approximately 4% higher than the random forest classifier. Additionally, Hossain et al. [9] suggested using the K-nearest neighbors classifier algorithm (KNN) for detecting and classifying plant leaf diseases. They extract texture parameter features from the diseased regions of leaf images and perform classification. Due to the imbalance between training time and final recognition accuracy in the K-nearest neighbors (KNN) classifier, Chanda et al. [10] addressed this issue by proposing the use of backpropagation for multi-class classification problems. By obtaining the weights within the neural network (NN) and applying particle swarm optimization (PSO) to automatically optimize the weights of the NN, they achieved a classification accuracy of 96%.

These models may have poor generalization and low universality on other datasets. With the continuous development of deep learning and the iterative updates of computer hardware, an increasing number of scholars are researching how to automatically recognize and classify images. LeCun et al. [11] made a groundbreaking contribution in 1998 by introducing the concept of convolutional neural networks (CNNs), which has become a milestone in the field of deep learning. Grinblat et al. [12] have demonstrated that CNNs outperform traditional machine learning algorithms in terms of plant identification, providing superior performance indicators. Ghosal et al. [13] proposed a method using CNN for identifying leaf diseases and extracting features. However, the accuracy of leaf classification for similar bacterial diseases was relatively low. T. Daniya and S. Vigneshwari [14] proposed a novel hybrid optimization algorithm for the classification of plant diseases. This method involves preprocessing through the extraction of regions of interest (ROI) and generating input images by combining datasets of both healthy and diseased rice plants. Employing a two-level classification approach, they differentiated between healthy and unhealthy categories, further subdividing the latter into three specific disease types. The achieved accuracy reached 0.9304, and an impressive F1 Score of 0.9142 was also attained. Sladojevic et al. [15] classified diseases in 13 categories of plant leaf regions using features extracted by CNN from images with a simple background. Sethy et al. [16] proposed combining SVM with CNN for the recognition of four types of diseases in 5932 rice images. This hybrid approach of integrating traditional machine learning with deep learning achieved high accuracy, but it suffered from the drawback of high time cost. Militante and Gerardo [17] utilized CNN for the recognition of healthy and diseased sugarcane images. While this method achieved high accuracy, it was noted that the background in the images was relatively uniform and simple. Brahimi et al. [18] proposed a variant of the GoogleNet network for the recognition of four different diseases in apple leaf regions, achieving an accuracy of 85.4%. When the background becomes more complex, researchers have proposed methods to improve the accuracy of the models. Li et al. [19] introduced the SE-Inception structure into the CNN framework, which enhances useful features and compresses redundant features, making the CNN model more versatile and robust. Gao et al. [20] incorporated the CA [21] attention mechanism into the base network, reducing the number of parameters and making the features more clear.

In recent years, Transformer [22] has made significant contributions to the field of natural language processing (NLP). It introduces innovative elements such as attention mechanisms, greatly enhancing the modeling and analytical capabilities of models for sequential data. However, one challenge is that Transformer models tend to be more complex. Han et al. [23] innovatively applied NLP encoders to the field of computer vision (CV). In the field of object detection, Carion et al. [24] employed a transformer-decoder for prediction, achieving higher accuracy compared to classical object detection frameworks. Ma et al. [25] proposed the need for lightweight network design. Lu et al. [26] highlighted that the decoding process in Transformer inevitably requires transforming features into one-dimensional vectors, which can be cumbersome and relatively inefficient.

The contributions of this paper are as follows:(1)The dataset used in this paper is relatively small, which is not suitable for deep CNN architectures. Therefore, we chose the ResNet-18 network with fewer layers. The residual modules in ResNet-18 address the issue of vanishing gradients and can effectively capture contextual information in the image, thus improving the model’s performance.(2)In each residual block of the CNN, we introduced the SE (squeeze-and-excitation) attention mechanism. This mechanism enhances the learning of inter-channel weights, adaptively analyzes the importance of certain channels, strengthens useful features, and reduces interference from irrelevant information and noise. This improves the model’s generalization ability.(3)After the CNN pooling layer, we incorporated the MHSA (multi-head self-attention) mechanism. This mechanism allows parallel computation of different input features, enabling each attention head to capture different disease image features. It helps extract spatial and channel relationships, resulting in better identification of disease texture and color features.(4)In the training set of the dataset, the threshold, K-means, and SVM algorithms are used to segment the lesions in the images. These segmented lesion images are then used to train the SE-VIT network to extract lesion features. Through comparative experiments, we found that using the SVM algorithm achieved the highest accuracy in disease spot extraction, reaching 89.57%. Furthermore, through heat map analysis, we observed that the regions of interest (ROIs) generated by the SVM algorithm were larger than those produced by the K-means and thresholding algorithms, indicating a better capture of disease spot information. Furthermore, in the heatmap, it can be observed that when using the threshold and K-means algorithms for feature extraction, the heat distribution focuses on areas other than the lesions. However, in the heatmap generated by SVM for lesion extraction, the color distribution primarily concentrates on the lesion areas, indicating better results. Therefore, the SVM algorithm for lesion extraction, combined with the network proposed in this paper, is more suitable for diagnosing diseases in sugarcane leaf parts in complex real-world environments.(5)This paper compares the SE-VIT model with the ResNet series of network models, the classic AlexNet network model, and the recent VisionTransformer network model, all of which achieve promising performance metrics. This highlights the advantages of the proposed model presented in this paper.(6)Finally, on the testing set consisting of four disease classes and healthy leaf, the addition of the SE attention mechanism increased the accuracy by 1.05%. Further incorporating an eight-head MHSA improved the accuracy by 3.01%. Lastly, the inclusion of 2D relative positional encoding further improved the accuracy by 1.04%. The final accuracy reached 89.57%, with a precision of 90.19%. Our proposed method also performed well on public datasets, demonstrating its effectiveness in reducing sugarcane yield losses. It exhibited good robustness and generalization, demonstrating its efficacy in mitigating sugarcane yield losses.

## 2. Materials

### 2.1. Data Source

In this paper, we selected two datasets to validate the performance of the proposed network model. One dataset is the publicly available PlantVillage dataset, and the other is a privately collected dataset called SLD. The PlantVillage dataset is primarily used for training the weights of the proposed hybrid network, SE-VIT. The training set in the private dataset SLD is used to train the SE-VIT network, while the test set is used to evaluate the network’s performance in complex real-world field backgrounds.

#### 2.1.1. PlantVillage

PlantVillage is an open-source public dataset of plant leaf diseases. The dataset consists of images depicting various plant leaf diseases and is known for having a relatively simple background. Prior to its public release, the dataset contained a total of 60,343 images, covering 14 common classes of plant leaf diseases with a total of 38 disease types. In Figure 1, we have presented partial disease samples on plant leaves: (a) displays apple scab, (b) showcases cherry powder mildew, (c) exhibits corn northern leaf blight, (d) demonstrates grape black rot, (e) illustrates peach bacteria spot, (f) displays pepper bacteria spot, (g) shows potato early blight, and (h) presents strawberry leaf scorch. Partial leaf disease images from the dataset are shown in Figure 1. To facilitate model training, the dataset is divided into a training set (42,240 images), a validation set (12,069 images), and a test set (6034 images). The dataset is available at https://data.mendeley.com/datasets/tywbtsjrjv/1, accessed on 1 September 2021.

#### 2.1.2. SLD

The data used in this study primarily pertain to the growth period of sugarcane, which typically lasts for 6 months. During the growth period, the stems and leaves of sugarcane grow rapidly, the leaves become lush, and sugar accumulates. It is during this stage that the incidence rate of sugarcane diseases is the highest, leading to a significant reduction in sugarcane yield. To validate the performance of the proposed hybrid model in complex environments, another dataset called SLD was selected in this study. The data for this dataset were collected from the agricultural cultivation and breeding base of Guangxi University. The images were captured using a Huawei Mate30 Pro smartphone with a resolution of 3648 × 2736 pixels. However, the designed network architecture in this paper was initially trained on a public dataset, where all images were resized to 224 × 224 pixels. The pre-trained weights from the public dataset were then transferred for fine-tuning on the training set of the private dataset. As a result, the images in the private dataset needed to be resized to 224 × 224 pixels to align with the network’s input requirements, and it also helped reduce computational resources, optimizing the model training process to a certain extent. The photos were taken at various times and from different angles. The SLD dataset consists of five categories of sugarcane leaf samples, including healthy leaves and four types of diseased leaves. Figure 2 illustrates examples of these categories, namely, ring spot disease, red stripe disease, brown stripe disease, and bacterial leaf spot. We did not consider using spectral methods such as fluorescence, FTIR, NIR, etc., to enhance the accuracy of our data collection during dataset collection. We plan to explore the use of these methods in future work on leaf disease identification.

The original dataset contained varying numbers of samples per category, ranging from 300 to 420 images, totaling 1877 original images. To address the issue of overfitting caused by imbalanced class distribution or insufficient samples in certain categories [27], all images were first resized to 224 × 224 pixels. Data augmentation techniques such as rotation and noise addition were then applied to enhance the dataset, resulting in a total of 8600 images. Table 1 shows the number of samples before and after augmentation of SLD.

### 2.2. Disease Spot Segmentation

The images of sugarcane leaf diseases are often captured in complex field backgrounds, which significantly affect the accuracy of disease recognition by the model. In this study, we extracted the regions of sugarcane leaf diseases from complex background images. SVM (support vector machine) is a traditional machine learning algorithm proposed by Vapnik and Corinna Cortes [28]. It is based on mathematical theory and can map linearly inseparable samples in a low-dimensional feature space to a high-dimensional space, enabling linear analysis of nonlinear samples. SVM is suitable for practical applications with small samples and complex nonlinearity. The extraction of disease regions from images is generally performed using unsupervised learning and supervised learning methods. Mortensen et al. [29] proposed 3D point cloud segmentation for estimating the fresh weight of lettuce, where dense point clouds were generated using a stereo camera by filtering out soil and weeds from the captured images. Reza et al. [30] combined the K-means clustering algorithm with the knowledge-based color and graph cut (KCG) algorithm to segment rice and weed regions using unmanned aerial vehicles for yield determination, but the segmentation results were not entirely satisfactory. Guijarro et al. [31] proposed a threshold segmentation method for segmenting the sky, soil, and plants. Due to the complex growth environment of sugarcane, it is difficult to achieve satisfactory segmentation results using simple unsupervised learning methods for extracting lesion regions from leaf images.

#### 2.2.1. Threshold

In this paper, we employed the OTSU algorithm, which is based on the idea of finding an optimal grayscale threshold to divide the image into two categories: foreground and background, such that the variance within each category is minimized, and the variance between the two categories is maximized. By doing so, the algorithm can automatically discover the most suitable threshold without the need for manual specification, thus achieving automatic image segmentation.
(1)σw^2(T)=w0(T)×σ0^2(T)+w1(T)×σ1^2T

In Equation (1), *w*0(*T*) and *w*1(*T*) represent the probabilities of foreground and background pixels, respectively, while *σ*0^2(*T*) and *σ*1^2(*T*) represent the grayscale variance of foreground and background pixels, respectively.
(2)σb^2(T)=w0(T)×w1(T)×(μ0(T)−μ1(T))^2

In Equation (2), *μ*0(*T*) and *μ*1(*T*) represent the mean grayscale values of foreground and background pixels, respectively. The optimal threshold T is found by selecting the grayscale level that maximizes the total variance *σw*^2(T) + *σb*^2(T). This threshold is considered the best threshold for image segmentation using the threshold algorithm.

#### 2.2.2. SVM

SVM is a commonly used machine learning algorithm for binary image classification. It aims to find a hyperplane that maximizes the margin (or boundary) between two different classes, effectively separating data points into two categories in such a way that the separating boundary is as far away as possible from the nearest data points. The training process of SVM can be viewed as a convex optimization problem, with the objective of maximizing the margin while satisfying accuracy requirements for classification. In two-dimensional space, a hyperplane can be represented by the linear Equation (3).
w × x + b = 0(3)

The distance from the hyperplane to the nearest data point is referred to as the margin. For binary classification problems, the margin of the hyperplane can be expressed as Equation (4), where ||w|| represents the norm (magnitude) of the normal vector w.
Margin = 2/||w||(4)

The training process of SVM can be viewed as a convex optimization problem, with the objective of finding the optimal hyperplane parameters w and b that maximize the margin while satisfying classification accuracy requirements. By solving this optimization problem, the optimal hyperplane can be obtained for classifying data points.

#### 2.2.3. K-Means

The K-means algorithm is a commonly used clustering algorithm that is used to divide data points into K distinct clusters. Subsequently, it calculates the distance between each data point and the centroids of each cluster, using the Euclidean distance as shown in Equation (5), In the formula, $x_{i}$ represents a data point, $c_{j}$ represents the cluster centroid, and n represents the number of features. For each data point $x_{i}$, select the cluster centroid $c_{j}$ that minimizes the $distance\left(x_{i},C_{j}\right)$, and assign $x_{i}$ to that cluster.
(5)$$distance\left(x_{i},C_{j}\right)=\sqrt{\Sigma_{k=1}^{n}{\left(x_{i},k-c_{j,k}\right)}^{2}}$$

With the K-means algorithm, data points can be divided into K distinct clusters, where data points within the same cluster have relatively small distances between them, while the distances between different clusters are relatively large. This achieves data clustering and analysis.

We finally applied the SVM algorithm to segment the diseased regions of sugarcane leaf images. In Figure 3, we compared the threshold algorithm, K-means clustering algorithm, and SVM. As shown clearly in Figure 3, the SVM algorithm was able to successfully segment the diseased regions, while the other algorithms either resulted in incomplete segmentation with missing lesion information or failed to segment the desired lesion regions. Therefore, in comparison to unsupervised learning-based segmentation algorithms, supervised learning-based image segmentation algorithms exhibited greater advantages. Consequently, we ultimately chose the SVM to achieve lesion segmentation in this study.

## 3. Methods

### 3.1. The Proposed SE-VIT

In this paper, we propose a hybrid neural network model called SE-VIT, which combines Transformer and CNN with SE attention modules for diagnosing sugarcane leaf diseases. The SE-VIT hybrid neural network model consists of three components: a ResNet-18 base network structure, SE attention mechanism modules, and an improved ViT module. The ResNet-18 network performs layer-wise feature extraction on the images, providing the extracted features for the subsequent attention mechanisms. The SE attention modules adjust the channel weights obtained from the base network, focusing on more important feature channels to improve the model’s generalization and robustness. The improved ViT module introduces 2D relative position encoding, which can be embedded after the convolutional layers of the CNN. It captures different feature relationships in different spatial and channel dimensions, enhancing the model’s expressive power. The technical roadmap and network structure of this paper are illustrated in Figure 4.

### 3.2. The SE-ResNet-18

SE-ResNet-18 is a CNN architecture designed for extracting global and channel features, inspired by ResNet-18 and the SE attention mechanism. It includes several 7 × 7 and 3 × 3 convolutional layers, max pooling layers, average pooling layers, fully connected layers, and SE attention modules. This hybrid network enhances the feature extraction process. The structure of the SE attention module is depicted in Figure 5. Each SE attention module is embedded after the last convolutional layer of each ResNet residual block, as shown in Figure 6. We refer to this module as the RSE module.

In Figure 5, the structure of the SE attention module is as follows: Firstly, the global pooling layer converts the feature map into a fixed-length vector, which helps extract global information. Then, through a series of fully connected layers, the features are linearly transformed using a weight matrix to achieve nonlinear combination. To address the issue of gradient vanishing and increase model sparsity, the ReLU (rectified linear unit) activation function is applied, which sets negative values to zero while leaving positive values unchanged. Sigmoid, as another activation function, transforms the input into a probability distribution. Finally, the input is normalized through the Scale operation to ensure appropriate weighting of the features. The SE attention module effectively captures and emphasizes important features by adaptively recalibrating feature responses.

SE-ResNet-18 consists of 8 RSE modules, along with pooling layers and convolutional layers, as shown in Figure 7. ResNet-18 was chosen because it has fewer layers compared to other ResNet series networks, making it easier to train. Since the dataset used in this study has a limited number of samples, the introduction of SE attention modules allows the network to adaptively adjust the channel weights and suppress irrelevant information similar to noise. This is particularly beneficial for small sample datasets. Additionally, the SE attention modules enhance the model’s expressive power and improve its classification performance. The detailed architecture of SE-ResNet-18 is illustrated in Figure 7.

### 3.3. Two-Dimensional MHSA

The original VIT module consists of absolute positional encoding, a series of multi-layer perceptrons (MLP), batch normalization layers, and multi-head self-attention (MHSA). The MLPs in the attention block allow for better feature learning. The batch normalization layer is a regularization technique that scales or shifts features to improve stability. In this study, the core module MHSA is selected. Through appropriate convolutional operations passed to the MHSA layer, a 1 × 1 convolutional layer is added to expand or reduce the dimensionality of the features before the MHSA layer. The inclusion of relative positional encoding aims to capture contextual information and focus more accurately on important features. The specific structure is illustrated in Figure 8.

MHSA (multi-head self-attention) is the core module that calculates different weights for the features obtained from previous layers. Each head calculates the correlations between different features from various perspectives, capturing feature relationships at different scales and angles to extract more diverse information. This facilitates the fusion of multiple heads. It can be computed using Equation (6).
(6)$$\mathrm{Attention}(Q,K,V)=\mathrm{Softmax}(\frac{QK^{T}}{\sqrt{d_{k}}})V$$

In Equation (1), the input features are used to generate different feature maps through three separate 1 × 1 convolutional layers, which are then used to generate Q, K, and V. The key step is to compute the correlation between Q and K. This is carried out by taking the dot product between Q and K, dividing it by the dimension of the features $d_{k}$, and then normalizing the weights using Softmax. These weights are then multiplied with the vector V through a dot product operation.

In MHSA, there is no explicit inclusion of positional information of the image. To address this, an embedded patches layer is introduced. This layer divides the input image into a large number of blocks and represents them as vectors of a certain length, effectively converting the two-dimensional vectors into one-dimensional vectors. The embedded patches layer takes advantage of the spatial correlations between the segmented image patches, enhancing the model’s ability to capture local information in the image.

In the classical VIT module, the positional encoding used before MHSA is the one-dimensional absolute positional encoding. There are three types of positional encoding in the encoder: non-positional, relative positional, and absolute positional encoding. Shaw et al. [32] conducted extensive experiments in CNN and RCNN models and demonstrated that relative positional encoding outperforms absolute positional encoding and non-positional encoding in self-attention mechanisms. Ramachandran et al. [33] introduced the concept of two-dimensional relative positional encoding, which helps preserve information between features and reduces information loss. The expression for two-dimensional relative positional encoding is shown in Equation (7).
(7)$$\mathrm{Attention}(Q,K,V)=\mathrm{Softmax}(\frac{QK^{T}+Q\;{\left(r_{i}+r_{j}\right)}^{T}}{\sqrt{d_{k}}})V$$

In Equation (2), $r_{i}$ and $r_{j}$ represent one-dimensional vectors of fixed width and height. Bello et al. [34] conducted extensive experiments and proposed the use of an eight-headed MHSA with relative positional encoding, which demonstrated good performance. It promotes the integration of information and avoids the limitations of having too few or too many heads. Fewer heads may not adequately capture complex features, while too many heads can make the model overly complex. Figure 9 illustrates a partial view of the relative positional encoding in the MHSA module.

## 4. Results

In this paper, all experiments were conducted on an Ubuntu environment using the Python 3.8 programming language. The experiments were performed on a workstation equipped with an Nvidia Geforce RTX 3090Ti, AMD (R) EPYC 7351P @ 2.4 GHz processor, 24 GB of graphics memory, and 16 cores. The deep learning framework version used was 1.10. The training was performed for 100 epochs, and the input image size was uniformly processed to 224 × 223 × 3. Following the initialization and learning rate proposed by He et al. [35], we set the learning rate to 0.0001. Additionally, we applied a cosine entropy decay learning strategy to ensure a smoother training process.

### 4.1. Model Evaluation Indices

In this text, to evaluate the performance of the model, various metrics were used to provide a comprehensive assessment. These metrics include accuracy, precision, recall, specificity, F1 score, Kappa score, and the number of model parameters. In the following equations, P (Positive) represents the total number of true positive instances, i.e., the total number of samples that belong to the positive class in the dataset. N (Negative) represents the total number of true negative instances, i.e., the total number of samples that belong to the negative class in the dataset. TP (True Positive) corresponds to the count of true positive instances, indicating the number of samples correctly predicted as positive by the model. TN (True Negative) corresponds to the count of true negative instances, indicating the number of samples correctly predicted as negative by the model. FP (False Positive) stands for false positive instances, which represents the number of samples incorrectly predicted as positive by the model. FN (False Negative) stands for false negative instances, indicating the number of samples incorrectly predicted as negative by the model. These metrics are used in classification problems to assess the performance of deep learning models and help measure the model’s ability to classify positive and negative instances.
(8)$$Accuracy=\frac{TP+TN}{P+N}$$
(9)$$Precision=\frac{TP}{TP+FP}$$
(10)$$Rcall=\frac{TP}{TP+FN}$$
(11)$$Specificity=\frac{TN}{FP+TN}$$
(12)$$F1\;score=\frac{2\times Precision\times Recall}{Precision+Recall}$$
(13)$$Kappa=\frac{P_{0}-P_{e}}{1-P_{e}}$$

### 4.2. Comparing Different Number of Heads in MHSA

In comparing different heads within MHSA on the SLD dataset, each head in MHSA can learn different weights, which aids the model in understanding different knowledge aspects. In this study, a series of experiments were conducted with varying numbers of heads to determine the optimal performance. Table 2 demonstrates that as the number of heads increased from 5 to 10, accuracy and other metrics consistently improved. However, when there were more heads, the metrics started to decline. This suggests that blindly stacking heads is not desirable as it introduces complex information that negatively impacts the results. Therefore, this paper selected eight heads as the optimal choice within MHSA.

### 4.3. Comparing Different Optimizers in SE-VIT

In deep learning, an optimizer is used to train neural network models. Its main purpose is to adjust the model’s parameters (weights and biases) to minimize the loss function. The optimizer’s goal is to find the best combination of model parameters so that the model can learn from the given training data and produce the best possible prediction results. As shown in Table 3, this paper compared two optimizers: stochastic gradient descent (SGD) and adaptive moment estimation (Adam) optimizer. The results indicated that the Adam optimizer outperformed the SGD optimizer in terms of accuracy, precision, and other metrics. Particularly, in Figure 10, it is evident that using the Adam optimizer resulted in faster convergence and higher accuracy compared to SGD. Based on these findings, in this study, we chose to use the Adam optimizer.

This paper ultimately chose eight heads for the multi-head self-attention (MHSA) module and selected the Adam optimizer. As shown in Table 4, the accuracy on the public dataset PlantVillage reached 97.26%, with a precision of 96.92%, recall of 96.68%, and an F1 Score of 0.968. These metrics demonstrate that the model proposed in this paper exhibits a significantly high level of generalization and accuracy. The training time for this model was 13.4 h. On the private dataset SLD, the accuracy achieved was 89.57%, precision was 90.19%, recall was 89.64%, and F1 Score was 0.896. The training time for this dataset was 3.1 h. Due to the large size of the dataset used in this article and the inclusion of a validation set, the risk of overfitting has been significantly reduced. Therefore, this article does not include additional discussions on cross-validation methods.

### 4.4. Ablation Experiments

#### 4.4.1. The Ablation Experiments on PlantVillage

Table 5 clearly presents the ablation experiments conducted on the PlantVillage public dataset. In this study, we initially experimented with the original CNN, which achieved an accuracy of 95.34%. Then, we incorporated the SE attention mechanism before the last convolutional layer of each residual block in the CNN. This selective weighting approach resulted in an improvement of 0.15% in accuracy. Next, we introduced MSHSA (multi-scale hierarchical self-attention), which further increased the accuracy by 1.69%. Finally, by incorporating relative position encoding to establish closer relationships between features, the accuracy reached 97.26%. Although the increase in model parameters, other performance metrics demonstrated various improvements, indicating that the proposed model in this paper exhibits strong generalization capabilities.

Figure 11 clearly demonstrates the process of accuracy changes during training on the PlantVillage dataset. It can be observed that the initial CNN model had a slow convergence rate, but after incorporating the SE attention mechanism, the convergence speed improved, resulting in a slight increase in accuracy. Furthermore, with the addition of the multi-head self-attention mechanism, another improvement in accuracy can be observed. Finally, after incorporating relative positional encoding, both the convergence speed and accuracy were further enhanced.

#### 4.4.2. The Ablation Experiments on SLD

Table 6 presents the results of the ablation experiments conducted on the SLD dataset. Initially, when using the traditional CNN, the accuracy obtained was 84.47%. After introducing channel attention, the accuracy improved to 85.52%. When incorporating MHSA (multi-head self-attention) and relative position encoding, the accuracy reached 88.53% and 89.57%, respectively. The inclusion of MHSA enables the extraction of different information features, while relative position encoding helps establish connections between pixels, resulting in improvements across various metrics.

### 4.5. Comparison of Different Networks

In order to demonstrate the advantages of our proposed model, we conducted comparisons with common classification models, including the classic AlexNet, the residual-based residual neural network (ResNet), and the recent vision transformer (ViT). The comparison results are shown in Table 7.

From the table above, it is evident that the SE-VIT model demonstrates excellent performance in terms of accuracy, precision, recall, and F1 score, with minimal variation in the number of samples per class. Comparing the residual-based residual neural network (ResNet), as the number of layers increases in ResNet-101 compared to ResNet-18, the model complexity increases, resulting in a 0.27% decrease in recall, an increased risk of false negatives, and a weakened ability of the classifier to identify all positive samples. In the classic AlexNet, the metrics show only marginal differences compared to ResNet-18, with a mere 0.004 improvement in the F1 score. Due to the small sample size of the dataset and the complex background information in the images, the vision transformer is sensitive to the position information within the images, leading to less accurate handling of position information. When compared to the SE-VIT model proposed in this paper, the vision transformer exhibits a 1.83% decrease in accuracy, a 1.71% decrease in precision, and a 1.97% decrease in recall, although its specificity metric, which represents the probability of correctly identifying all negative examples, is 0.57% higher than that of the SE-VIT model. Finally, in terms of the F1 Score metric, the SE-VIT model outperforms the vision transformer model by 1.5%. Considering all metrics, high accuracy alone is not sufficient; high precision and recall are also crucial. In terms of the speed metric, frames per second (FPS), the model proposed in this paper is lower than AlexNet. However, it outperforms AlexNet in accuracy by 4.52%, precision by 3.36%, and recall by 4.49%. The performance of the proposed model is better than that of AlexNet. Finally, this paper also compared the Swin-transformer network architecture. The SE-VIT network achieved an improvement of 0.81% in accuracy, 0.84% in precision, 1.29% in recall, and 0.8% in F1 score when compared to the Swin-transformer network. Although the FPS (frames per second) is three lower than the latest model, we consider it fully acceptable given the overall performance improvement. Therefore, in real-world complex environments, the SE-VIT model proposed in this paper achieves a balanced performance in accuracy, recall, precision, and F1 score, meeting the requirements for detecting sugarcane leaf disease in this project.

### 4.6. Comparison of the Final Performance of Different Disease Segmentation Methods

In the study, three different segmentation methods were employed to extract disease lesions on sugarcane leaf surfaces. These methods include the threshold algorithm, K-means algorithm, and SVM algorithm. As shown in Figure 3, a comparison of the algorithms reveals that the SVM algorithm captures more distinctive features in the extracted lesions compared to the other two methods. Figure 12 presents the confusion matrices obtained by the three lesion extraction algorithms.

In Figure 13, we compared the accuracy of five different categories obtained by applying different lesion algorithms on the test set. It is evident that SVM consistently performs well with high accuracy. It can be observed that while SVM may have lower performance in specific disease recognition, it exhibits excellent overall performance across multiple metrics. Therefore, choosing the SVM algorithm as the most suitable method for lesion extraction is preferred.

Different lesion extraction algorithms employ different focusing mechanisms, which can introduce errors when analyzing images. The region of interest (ROI) becomes crucial in this context. To gain an intuitive understanding, we generated heatmaps from the last layer of the model. Figure 14 displays the heatmaps generated by different lesion extraction methods. It is clear from the image that the lesions segmented by the SVM algorithm are more easily captured by the model compared to the threshold and K-means algorithms. In the figure, it can be observed that the heat map in the original image does not accurately capture the characteristics of the lesion in the hotspot area. Due to the higher accuracy of the SVM algorithm in segmenting lesions compared to the other two algorithms, the heatmaps visualizing these features reveal that the heatmap of the lesion region can provide more precise localization, thereby aiding the model in better identifying the diseased areas.

In Figure 15, the accuracy of disease classification using different lesion extraction algorithms in a real-world environment is demonstrated. In this figure, “Bac” represents Bacterial, “BS” represents Brown Stripe, “RdS” represents Red Stripe, “RgS” represents Ring Spot, and “prob” represents probability. It can be observed from the figure that when the SVM algorithm is used for lesion extraction, the accuracy of disease classification for various sugarcane leaf parts is above 85% in all cases. Compared to the algorithms of threshold and K-means for lesion extraction, there is a significant improvement in accuracy. It is clear that when the threshold algorithm is used, the final detection and classification accuracy is only around 60%, making it unsuitable for use in complex real-world environments. The K-means algorithm shows better performance compared to the threshold algorithm, but there is still a noticeable gap compared to the SVM algorithm in terms of overall performance. Furthermore, in the heatmap, it can be observed that when using the threshold and K-means algorithms for feature extraction, the heat distribution focuses on areas other than the lesions. However, in the heatmap generated by SVM for lesion extraction, the color distribution primarily concentrates on the lesion areas, indicating better results. Therefore, the SVM algorithm for lesion extraction, combined with the network proposed in this paper, is more suitable for diagnosing diseases in sugarcane leaf parts in complex real-world environments.

In addition, this paper classifies diseases in four different sugarcane leaf parts under low-light conditions. Figure 16 showcases the outstanding performance achieved by the SE-VIT network proposed in this paper, combined with SVM for lesion extraction and final classification. This demonstrates that the network proposed in this paper, when combined with SVM, delivers exceptional performance even under low-light conditions.

## 5. Discussion and Conclusions

This study contributes to the broader body of research on the intersection of machine learning and diseases in tropical crops. The utilization of attention mechanisms in the SE-VIT hybrid network showcases the continuous advancement of artificial intelligence techniques in agriculture. It aligns with the growing trend of harnessing the potential of machine learning to address agricultural challenges worldwide. The findings and methodology from this study may also be adapted for diagnosing diseases in other tropical crops [36,37], widening the scope of its applicability [38,39]. By linking this research with other studies in the field, a holistic understanding of the role of machine learning such as random forest (RF), the debiased sparse partial correlation (DSPC) algorithm, and support vector machine (SVM) in disease diagnosis and management in tropical agriculture can be achieved [40,41,42], facilitating the development of innovative and sustainable solutions for farmers in these regions.

In this paper, based on a small-scale dataset, we selected a CNN with fewer layers, specifically ResNet18. Within each residual block, we incorporated the SE attention module after the last convolutional layer. This module helps reduce the interference of irrelevant information and allows the model to adaptively learn different weights, thereby reducing unnecessary information. Additionally, we leveraged the MHSA (multi-head self-attention) component, a core element of the VIT module. Through experimentation, we found that using eight heads yielded the best results. We further introduced the relative position encoding module to form the new 2D-MHSA module. By incorporating relative position encoding, the model gains a better understanding of the spatial relationships between image pixels and mitigates the negative impact of positional biases. Through ablation experiments conducted on both public and private datasets, we observed accuracy improvement of 5.1% and 1.92% as well as precision improvement of 1.94% and 3.23%. Compared to other models, SE-VIT has achieved excellent performance in metrics such as accuracy, precision, recall, and F1 score.

In order to eliminate complex backgrounds and achieve accurate recognition by the model, we compared different image processing techniques for lesion extraction, including original images, K-means algorithm, threshold algorithm, and SVM algorithm. Through experimentation, it was verified that the ROI regions generated by the SVM algorithm were larger than those obtained by the K-means and threshold algorithms. Therefore, combining SVM with the proposed SE-VIT model yielded the best results.

## Figures and Tables

**Figure 1 sensors-23-08529-f001:**
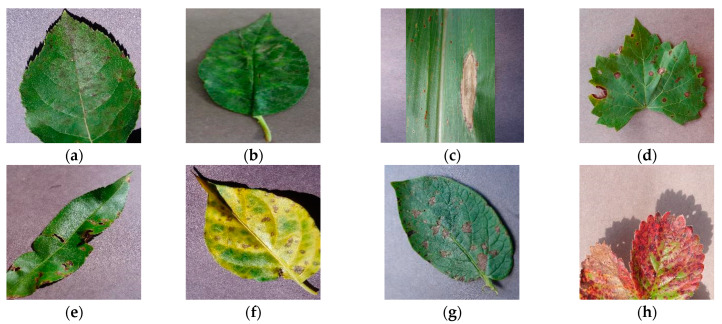
Examples of PlantVillage disease dataset images with simple backgrounds; (**a**) Apple: scab; (**b**) Cherry: powder mildew; (**c**) Corn: northern leaf blight; (**d**) Grape: black rot; (**e**) Peach: bacteria spot; (**f**) Pepper: bacteria spot; (**g**) Potato: early blight; (**h**) Strawberry: leaf scorch.

**Figure 2 sensors-23-08529-f002:**
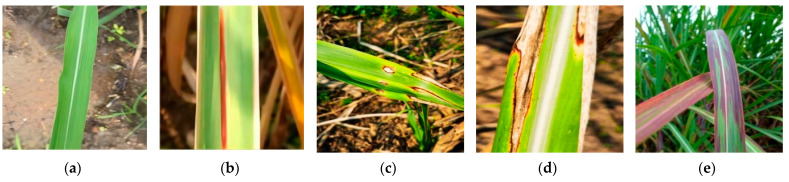
Four sugarcane leaf diseases and healthy leaf; (**a**) Healthy; (**b**) RedStripe; (**c**) RingSpot; (**d**) BrownStripe; (**e**) Bacterial.

**Figure 3 sensors-23-08529-f003:**
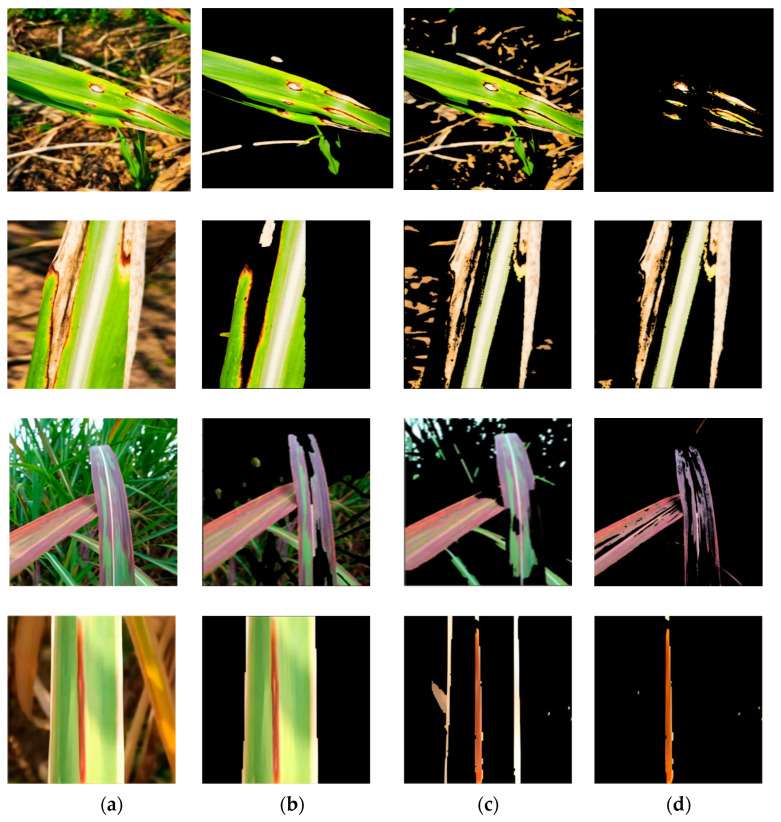
Disease spot segmentation results. (**a**) the original images; (**b**) threshold segmentation results; (**c**) K-means clustering results; (**d**) SVM results.

**Figure 4 sensors-23-08529-f004:**
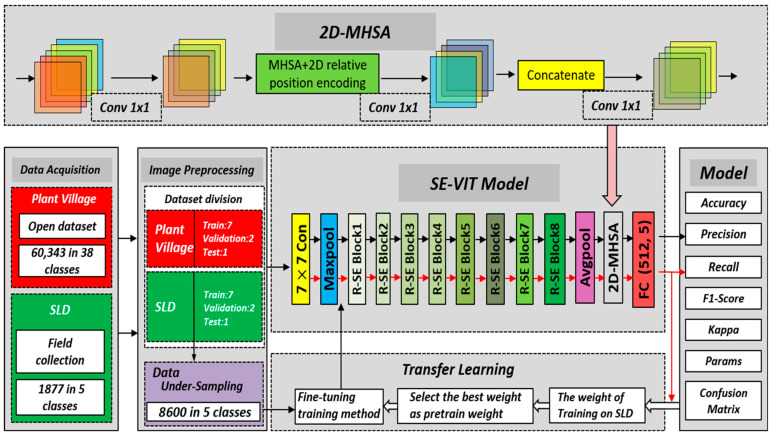
The technology roadmap of this study and SE-VIT architecture.

**Figure 5 sensors-23-08529-f005:**
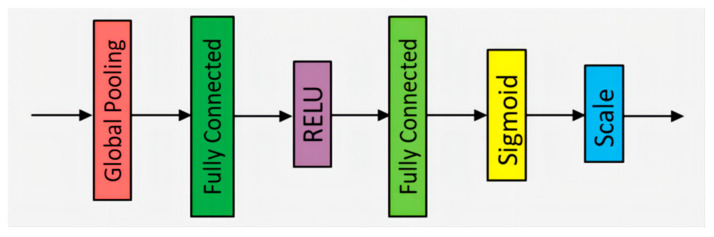
The structure of the SE attention module.

**Figure 6 sensors-23-08529-f006:**
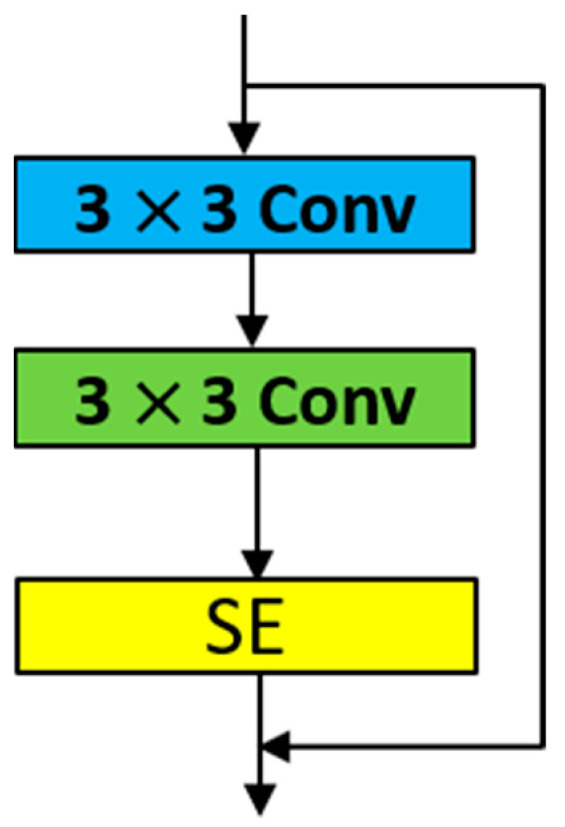
SE attention embedded into the residual block.

**Figure 7 sensors-23-08529-f007:**
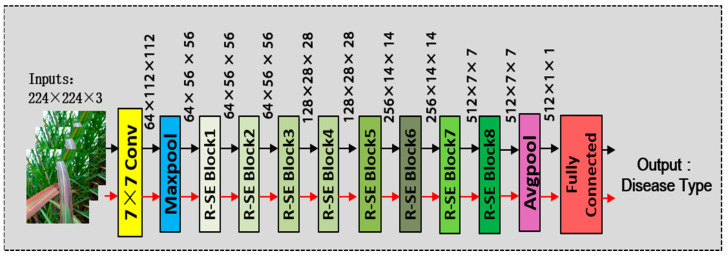
The structure of the SE-ResNet-18.

**Figure 8 sensors-23-08529-f008:**
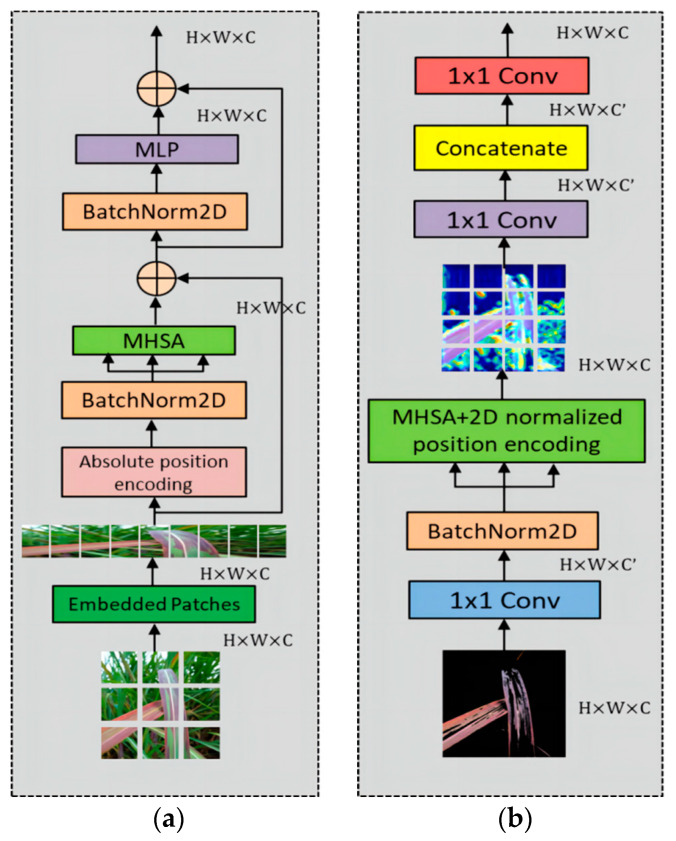
The structure of (**a**) ViT and (**b**) 2D-MHSA block.

**Figure 9 sensors-23-08529-f009:**
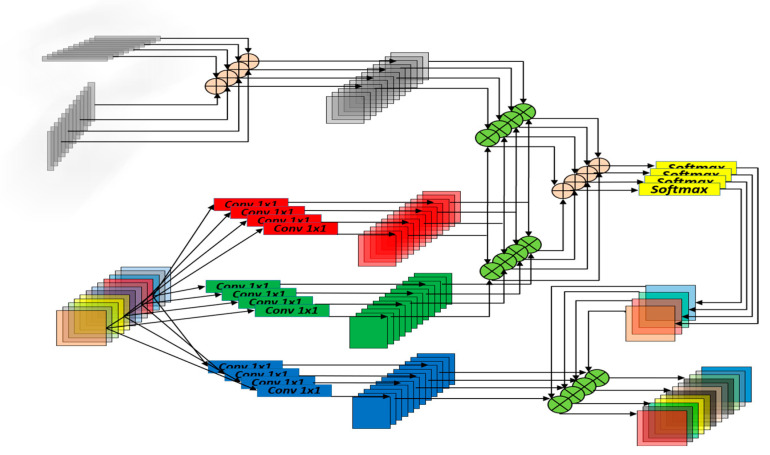
Schematic diagram of the MHSA mechanism.

**Figure 10 sensors-23-08529-f010:**
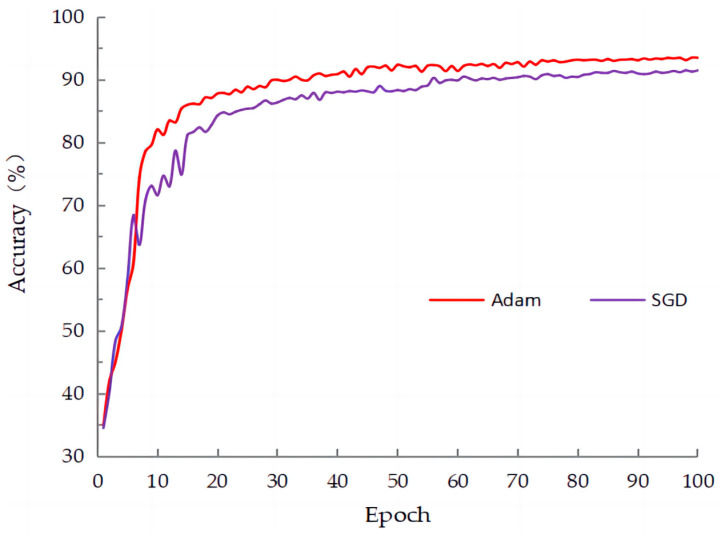
The training curves of the two optimizers on SLD.

**Figure 11 sensors-23-08529-f011:**
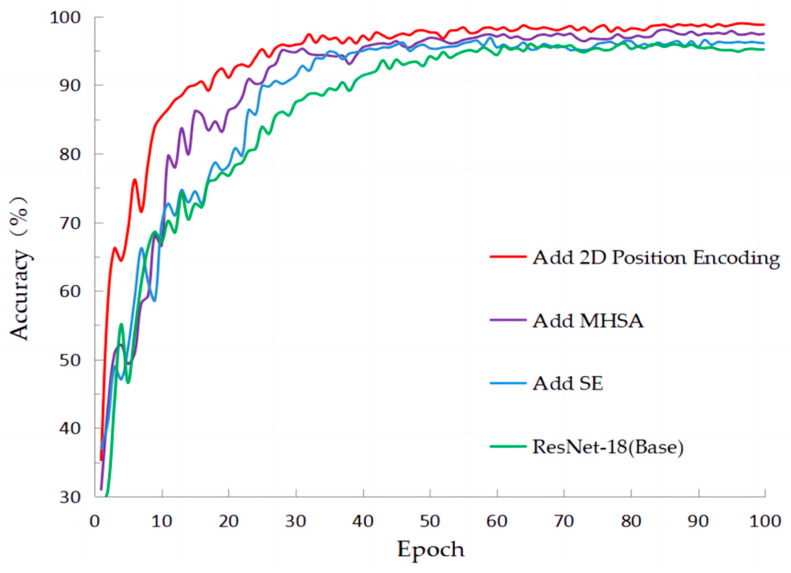
The accuracy of training in classification with ablation experiments on PlantVillage.

**Figure 12 sensors-23-08529-f012:**
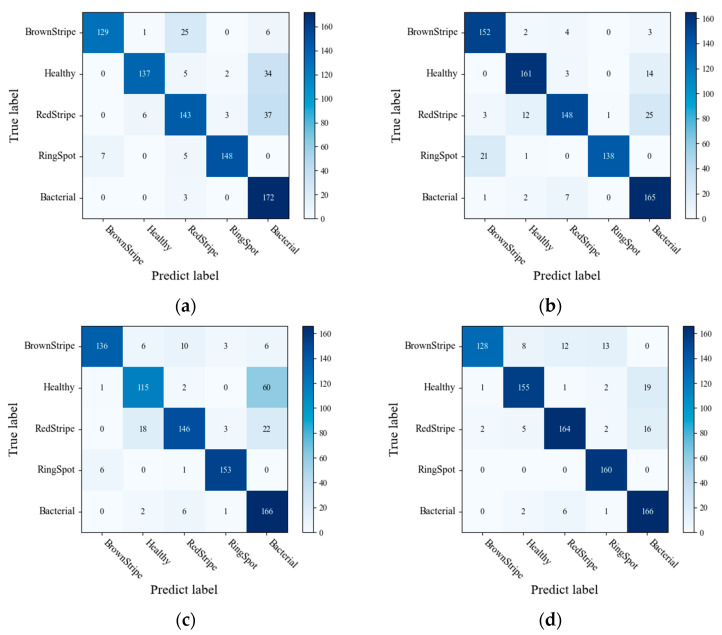
Confusion matrix for (**a**) Original (**b**) K-means (**c**) Threshold (**d**) SVM.

**Figure 13 sensors-23-08529-f013:**
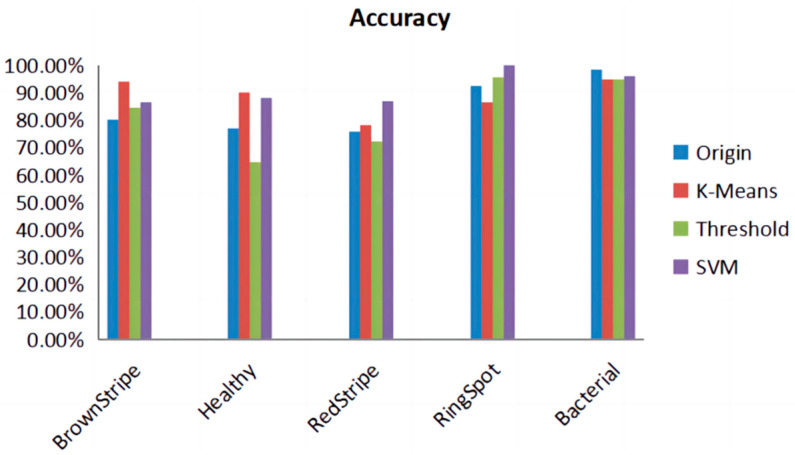
Accuracy of lesion extraction algorithms for different diseases. It is evident from Table 1, showing the sample sizes before and after augmentation of SLD, that the model, when combined with the SVM image processing, achieves an accuracy of 89.57%, precision of 90.19%, recall of 89.64%, specificity of 97.38%, and an F1 score of 0.896. The comprehensive metrics indicate that, following the lesion extraction stage and in conjunction with the SE-VIT network, a notable classification performance is achieved. See Table 8.

**Figure 14 sensors-23-08529-f014:**
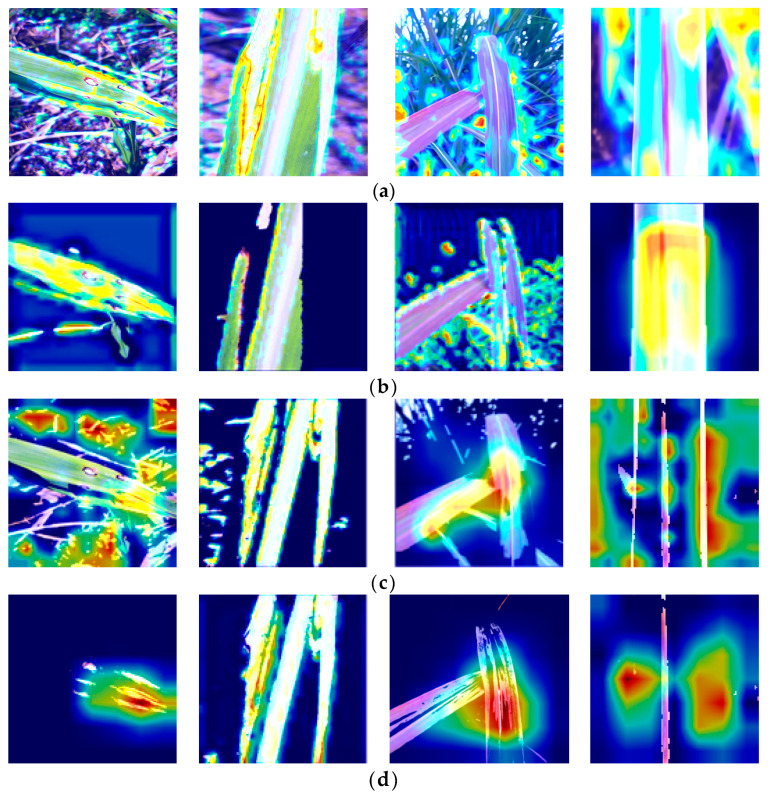
Heatmaps obtained by different lesion extraction algorithms; (**a**) Original; (**b**) Threshold; (**c**) K-means; (**d**) SVM.

**Figure 15 sensors-23-08529-f015:**
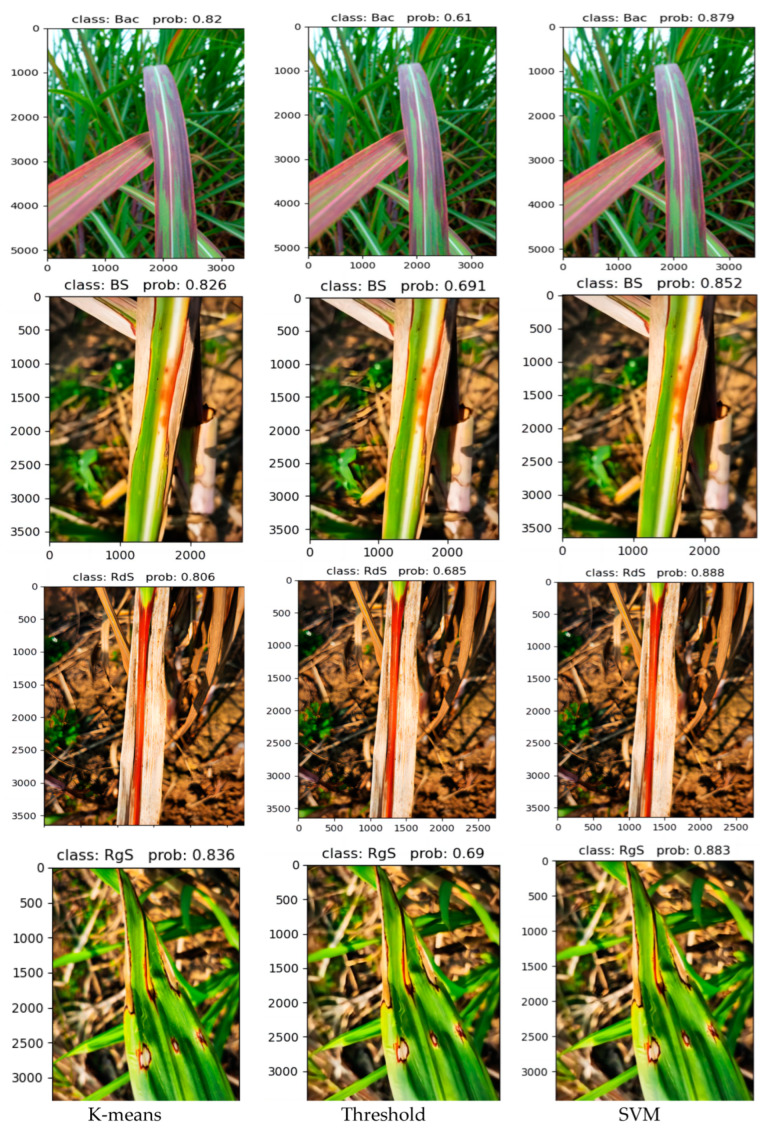
Different algorithms extract the final classification results of lesions.

**Figure 16 sensors-23-08529-f016:**

Performance under low-light conditions.

**Table 1 sensors-23-08529-t001:** The sample sizes before and after augmentation of SLD.

Classes	Original				Data Augmentation			
	Total	Training	Validation	Testing	Total	Training	Validation	Testing
Healthy	340	238	68	34	1780	1246	356	178
RedStripe	414	290	83	41	1890	1323	378	189
RingSpot	365	255	73	37	1600	1120	320	160
BrownStripe	370	259	74	37	1610	1127	322	161
Bacteria	388	269	80	39	1750	1225	350	175
Total	1877	1311	378	188	8600	6041	1726	863

**Table 2 sensors-23-08529-t002:** Comparison of the different heads in MHSA on SLD.

Number of Heads in MHSA	Accuracy /%	Precision /%	Recall /%	Specificity /%	F1 Score
2	82.97	84.30	83.21	95.72	0.833
4	87.49	89.29	87.70	96.83	0.874
8	89.57	90.19	89.64	97.38	0.896
10	84.71	86.40	84.64	96.16	0.848

**Table 3 sensors-23-08529-t003:** Comparison of different optimizers in SE-VIT on SLD.

Different Optimizers	Accuracy /%	Precision /%	Recall /%	Specificity /%	F1 Score
SGD	87.61	88.16	87.81	96.87	0.879
Adam	89.57	90.19	89.64	97.38	0.896

**Table 4 sensors-23-08529-t004:** Performance on SLD and PlantVillage.

Dataset	Accuracy /%	Precision /%	Recall /%	Specificity /%	F1 Score	Training Time
SLD	89.57	90.19	89.64	97.38	0.896	3.1 h
PlantVillage	97.26	96.92	96.68	99.94	0.968	13.4 h

**Table 5 sensors-23-08529-t005:** The ablation experiments on PlantVillage.

Trial	Accuracy /%	Precision /%	Recall /%	Specificity /%	F1 Score	Parameters ×10^6^
ResNet18 (Baseline)	95.34	94.98	92.49	99.98	0.931	11.20
+SE	95.49	94.65	92.67	99.89	0.934	11.69
+MHSA	97.18	96.79	96.28	99.92	0.965	12.97
+2D Position Encoding	97.26	96.92	96.68	99.94	0.968	13.55

**Table 6 sensors-23-08529-t006:** The ablation experiments on SLD.

Trial	Accuracy /%	Precision /%	Recall /%	Specificity /%	F1 Score	Parameters ×10^6^
ResNet18 (Baseline)	84.47	86.96	84.47	96.88	0.850	11.18
+SE	85.52	86.66	85.83	96.36	0.859	11.26
+MHSA	88.53	89.33	88.74	97.13	0.887	12.38
+2D Position Encoding	89.57	90.19	89.64	97.38	0.896	13.26

**Table 7 sensors-23-08529-t007:** Network comparison table.

Model	Accuracy/%	Precision/%	Recall/%	Specificity/%	F1 Score	FPS
ResNet-18	84.47	86.96	84.47	96.88	0.850	80
ResNet-101	84.49	84.77	84.20	97.24	0.846	45
AlexNet	85.05	86.83	85.15	96.23	0.854	96
Vit-16	87.74	88.48	87.67	97.95	0.881	82
Swin-Transformer	88.76	89.35	88.96	97.18	0.889	89
SE-VIT(ours)	89.57	90.19	89.64	97.38	0.896	86

**Table 8 sensors-23-08529-t008:** After using the SVM segmentation algorithm, the performance metrics of SE-VIT.

Accuracy/%	Precision/%	Recall/%	Specificity/%	F1 Score
89.57	90.19	89.64	97.38	0.896

## Data Availability

The PlantVillage dataset is available at: https://data.mendeley.com/datasets/tywbtsjrjv/1, accessed on 1 September 2021. Due to limitations in geographic location and climate, the captured images may not comprehensively cover various stages of diseases. In our upcoming work, we are planning to extensively study diseases at different stages and document them through photographs to enrich our dataset. Given that the SLD data used in this study were self-collected, the dataset is being further improved. The dataset used in this study only includes several common diseases during the sugarcane growth period. Our future research will further enrich the dataset of diseases affecting various parts of sugarcane leaves, including those that are less common. Thus, the private dataset is unavailable at present.

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
