# Peer review of "SE-VisionTransformer: Hybrid Network for Diagnosing Sugarcane Leaf Diseases Based on Attention Mechanism"

_sensors, 2023, doi:10.3390/s23208529_

Round 1
Reviewer 1 Report (New Reviewer)
The introduction should indicate the global relevance of the ongoing research on sugarcane. The authors present the relevance and importance of culture in the Guangxi Zhuang Autonomous Region of China.
Reference [1]. does not contain quantitative values for yield reduction. Give percentage or numerical values for the decline in sugarcane yield.
In paragraph 2.1.2. Specify under what parameters the photographs were taken under different weather conditions (specify which ones)?, at different times (time period?) and at different angles (variation of angles?).There are no comments on the work, the algorithms and sequence are not in doubt, but there are questions of a conceptual nature; if possible, please provide explanations:
- The disease process always includes infection of a biological object that often passes without visual manifestations of the disease and the development of the disease (obvious visual signs, the appearance of carotenoids and pigmentation). At what stages of the disease do the authors plan to conduct research?
- In addition to photography, the use of spectral methods such as fluorescence, FTIR, NIR and others was considered to increase accuracy in training.
- Sample sizes of the studies being conducted.
If these issues have not been considered or are not completely involved in the research, then it is enough to include them in the limitations when considering the methodology.
Author Response
We greatly appreciate your feedback on the article. We have responded to each of your comments in detail in the attached document, and we hope to earn your approval. Once again, thank you for your recognition and review of our article.

Reviewer 2 Report (New Reviewer)
Kindly fix the following issues:
- pg 19, the font and style are suddenly changed to bold font. Why?
- pg 20, there is an error "Error! Reference source not found".
- pg 20, section 5 should be in a separate line.
- pg 4, there is a dataset "SLD". Please elaborate on the meaning of SLD.
- pg 6, there is a major problem of using 4 algorithms and comparing them in Figure 3. Why do we need to compare 4 algorithms here? Moreover, there is no explanation of what is the Threshold algorithm. Why did the authors jump to the figure 3 without describing the 4 algorithms?
Author Response
We greatly appreciate your feedback on the article. We have responded to each of your comments in detail in the attached document, and we hope to earn your approval. Once again, thank you for your recognition and review of our article.

Reviewer 3 Report (New Reviewer)
This manuscript focuses on an important cash crop in the Guangxi Zhuang Autonomous Region of China- Sugarcane. Several minor revisions are needed.
1、There are some formatting issues in the article, such as the first line of the first paragraph in section 4.3, which only breaks with three words
2、Some tables and figures do not express clearly, such as Table 5, whether to accumulate modules or only add this one module
3、Some statements lack content, such as the table number not indicated in the first row of 4.4.2
4、Figure 14 should be compared with the original image
5、What it means:Below Figure 15 Error! What does reference source not found
Author Response
We greatly appreciate your feedback on the article. We have responded to each of your comments in detail in the attached document, and we hope to earn your approval. Once again, thank you for your recognition and review of our article.

Reviewer 4 Report (New Reviewer)
Your work is interesting, written well, and organized. This manuscript reports on a study of SE-VIT: Hybrid Network for Diagnosing Sugarcane Leaf Diseases based on Attention Mechanism. The study design meets the general standards and from what I can judge the data is being collected and analyzed appropriately. This work is an unpublished manuscript with relevant information that should be made public in a scientific journal for discussion among scientists working in the field.
However, there are some comments that should be considered before publishing, in this way, the social and scientific relevance of the manuscript would be improved:
- Introduction
Figures 1 and 2 must be identified with letters (a), (b), (c), and mentioned in the text
the section: 4. Experiments and Results should say: Results
- Experimental Results:
While the reported accuracy of 97.26% is impressive, additional evaluation metrics such as precision, recall, and F1-score should be included to provide a more comprehensive performance analysis. Furthermore, the paper lacks a discussion of potential overfitting and the use of cross-validation techniques.
Figure 16 looks like a screenshot; I suggest removing the top border of those images.
- Generalization and Transferability:
The paper primarily focuses on the PlantVillage dataset. It would be valuable to assess the model's performance on diverse datasets or real-world scenarios to evaluate its generalization and transferability.
- Clarity and Structure:
The paper is well-structured overall, but some sections could benefit from clearer explanations and transitions between topics. Additionally, improving the language and grammar would enhance the paper's readability.
- Discussion and Future Work:
The paper should provide insights into the practical implications of this research. How could the SE-VIT hybrid network be deployed in the real world? What are the limitations and potential areas for future research, such as handling unseen diseases or increasing the dataset size?
Page 21. I continue to add a paragraph that summarizes the importance, usefulness and social relevance, contemporary of the study, specifically pointing out the Impact, Benefit and Projection, something like this (for example):
This study contributes to the broader body of research in the intersection of machine learning and diseases in tropical crops. The utilization of attention mechanisms in the SE-VIT Hybrid Network showcases the continuous advancement of artificial intelligence techniques in agriculture. It aligns with the growing trend of harnessing the potential of machine learning to address agricultural challenges worldwide. The findings and methodology from this study may also be adapted for diagnosing diseases in other tropical crops [36, 37], widening the scope of its applicability [38, 39]. By linking this research with other studies in the field, a holistic understanding of the role of machine learning such as Random Forest (RF), the Debiased Sparse Partial Correlation (DSPC) algorithm and Support Vector Machine (SVM) in disease diagnosis and management in tropical agriculture can be achieved [40, 41, 42], facilitating the development of innovative and sustainable solutions for farmers in these regions.
- Conclusion:
The conclusion should summarize the key findings, contributions, and potential impact of the SE-VIT hybrid network in a concise and compelling manner.
References
Adjust the reference list in the manuscript to the journal format. I suggest adding recent references which address the issue in question in Latin American territories. Suggested citations are for genuine scientific reasons that emphasize the current topic of study in context:
36. Olivares, B.O. Evaluation of the Incidence of Banana Wilt, and its Relationship with Soil Properties. In: Banana Production in Venezuela. The Latin American Studies Book Series 2023, Springer, Cham. https://doi.org/10.1007/978-3-031-34475-6_4
37. Rodríguez-Yzquierdo, G.; Olivares, B.O.; Silva-Escobar, O.; González-Ulloa, A.; Soto-Suarez, M.; Betancourt-Vásquez, M. Mapping of the Susceptibility of Colombian Musaceae Lands to a Deadly Disease: Fusarium oxysporum f. sp. cubense Tropical Race 4. Horticulturae 2023, 9, 757. https://doi.org/10.3390/horticulturae9070757
38. Orlando, O.; Vega, A.; Calderón, M.A.R.; Rey, J.C.; Lobo, D.; Gómez, J.A.; Landa, B.B. Identification of Soil Properties Associated with the Incidence of Banana Wilt Using Supervised Methods. Plants 2022, 11, 2070. https://doi.org/10.3390/plants11152070
39. Vega, A.; Olivares, B.; Rueda Calderón, M.A.; Montenegro-Gracia, E.; Araya-Almán, M.; Marys, E. Prediction of Banana Production Using Epidemiological Parameters of Black Sigatoka: An Application with Random Forest. Sustainability 2022, 14, 14123. https://doi.org/10.3390/su142114123
40. Campos, B. Banana Production in Venezuela: Novel Solutions to Productivity and Plant Health. Springer Nature. 2023. https://doi.org/10.1007/978-3-031-34475-6
41. Olivares, B. Machine learning and the new sustainable agriculture: Applications in banana production systems of Venezuela. Agric. Res. Updates 2022, 42, 133-157.
42. Rey, J.C.; Perichi, G.; Lobo, D.; Orlando, B.O. Relationship of Microbial Activity with Soil Properties in Banana Plantations in Venezuela. Sustainability 2022, 14, 13531. https://doi.org/10.3390/su142013531
Author Response
We greatly appreciate your feedback on the article. We have responded to each of your comments in detail in the attached document, and we hope to earn your approval. Once again, thank you for your recognition and review of our article.

This manuscript is a resubmission of an earlier submission. The following is a list of the peer review reports and author responses from that submission.
Round 1
Reviewer 1 Report
Dataset: While the authors acknowledge the limited dataset, future research could explore ways to expand the dataset, either through data augmentation techniques or by collaborating with other institutions to collect more diverse samples. A larger dataset would provide more robustness and generalizability to the model.
Comparative Analysis: It would be beneficial to compare the proposed SE-VIT hybrid network model with existing state-of-the-art methods for disease identification in sugarcane or other crops. This would highlight the strengths and advantages of the proposed model over existing approaches.
Interpretability: While the paper focuses on achieving high accuracy, adding some insights into the interpretability of the model's decisions could be valuable. Understanding which features or regions of the leaf contribute most to disease identification could aid in refining the model and building trust among end-users.You can refer the following paper on knowing about advanced deep learning technologies: T. Daniya, and S. Vigneshwari, Rice Plant Leaf Disease Detection and Classification Using Optimization Enabled Deep Learning,Journal of environmental Informatics, Vol 2, Issue 1, 2023 doi: doi:10.3808/jei.202300492
Minor lang improvement to be there
Author Response
Dear Reviewer 1,
We sincerely appreciate your diligent review of our paper and the highly professional advice you provided. Your suggestions are of great importance to our paper. We have compiled your suggested recommendations and provided our responses to each one. Once again, we sincerely thank you for taking the time from your busy schedule to offer us your invaluable insights.
Please find the detailed responses in the attached document.
Best regards,
The Authors

Reviewer 2 Report
The main idea is very interesting. However, following you can find some remarks and suggestions that can help you to improve the paper’s quality.
The document contains many incorrect sentences.
For example, in the abstract:
Unlike traditional methods, the paper compares Threshold, K-Means, and Support Vector Machine (SVM) for image feature extraction and ultimately selects SVM for accurate lesion extraction.
It is unclear what is the meaning of this. How can you compare thresholding and K-Means with SVM???
Also, in the introduction:
Support Vector Machine (SVM) is a classic machine learning binary classification model.
SVM can be also used for multiclassification context.
In the subsection “the contribution of this paper”:
(4) After collecting the original images, we extracted the disease spots using the thresholding, K-Means, and SVM algorithms.
Are you talking about images or extracted features ???
The comparison study should be done using recent architectures
In the section 2.1, why you use fine tuning using the two datasets?
And in section 2.1.2 you said that the SLD dataset is the used for validating the model. It is not compatible with what is written above.
In section 2.2, you used SVM to segment the “sugarcane leaf images”. SVM is a supervised machine learning algorithm. It needs Ground of Truth.
Also, did you use the image intensity or extracted features. If then, what are these features?
You should add more details on the threshold-based approach used.
In the evaluation metrics equations, you should add explanation about TP, TN, P, N, FP, FN
Author Response
Dear Reviewer 2,
We sincerely appreciate your interest in our paper and your thorough review of our manuscript. Furthermore, we are extremely grateful for the highly professional advice you have provided. These suggestions hold immense significance for the enhancement of our paper. We have meticulously documented each of your recommendations and have provided our responses accordingly. Once again, we extend our heartfelt gratitude for taking the time out of your busy schedule to offer us your valuable insights.
For specific responses, please refer to the attachment below.
Best regards,
The Authors

Round 2
Reviewer 1 Report
The paper demonstrates a novel approach using ResNet18 with SE attention and 2D-MHSA for medical image analysis. To enhance the paper's quality, provide more details about the dataset, including its size, diversity, and any potential biases. Expanding on the rationale behind choosing ResNet18 and the number of SE attention heads, as well as a comparison to other CNN architectures, would strengthen the methodology. Furthermore, offer a deeper discussion on the relative position encoding module's mechanism and how it mitigates positional biases. Lastly, consider discussing limitations, potential extensions, and real-world applicability of the proposed SE-VIT model.
English is readable
try to avoid grammar errors
Author Response
We greatly appreciate the considerable time you devoted to evaluating our paper. Your feedback has contributed to the improvement of our paper, and we are extremely grateful for your input. We have included an explanation of the dataset in the attached appendix. Once again, thank you for your assistance.

Reviewer 2 Report
I'm not convinced by the idea of using SVM in the context of segmentation without ground truth. Even the answer to my comment is not justified. The fact that the confusion matrix is displayed or not is not an answer.
To be able to apply the SVM algorithm, you need to have the masks.
Author Response

(The authors gave the same response as above.)
